# Immune Modulatory Effects of Vitamin D on Viral Infections

**DOI:** 10.3390/nu12092879

**Published:** 2020-09-21

**Authors:** Maheen Siddiqui, Judhell S. Manansala, Hana A. Abdulrahman, Gheyath K. Nasrallah, Maria K. Smatti, Nadin Younes, Asmaa A. Althani, Hadi M. Yassine

**Affiliations:** 1College of Health Science-QU Health, Qatar University, Doha 2713, Qatar; ms1308694@student.qu.edu.qa (M.S.); jm1403146@student.qu.edu.qa (J.S.M.); gheyath.nasrallah@qu.edu.qa (G.K.N.); nyounes@qu.edu.qa (N.Y.); aaja@qu.edu.qa (A.A.A.); 2Biomedical Research Center, Qatar University, Doha 2713, Qatar; ha-797-ha@hotmail.com (H.A.A.); msmatti@qu.edu.qa (M.K.S.)

**Keywords:** vitamin D, deficiency, hepatitis, Influenza, Covid-19, AIDS

## Abstract

Viral infections have been a cause of mortality for several centuries and continue to endanger the lives of many, specifically of the younger population. Vitamin D has long been recognized as a crucial element to the skeletal system in the human body. Recent evidence has indicated that vitamin D also plays an essential role in the immune response against viral infections and suggested that vitamin D deficiency increases susceptibility to viral infections as well as the risk of recurrent infections. For instance, low serum vitamin D levels were linked to increased occurrence of high burdens viral diseases such as hepatitis, influenza, Covid-19, and AIDS. As immune cells in infected patients are responsive to the ameliorative effects of vitamin D, the beneficial effects of supplementing vitamin D-deficient individuals with an infectious disease may extend beyond the impact on bone and calcium homeostasis. Even though numerous studies have highlighted the effect of vitamin D on the immune cells, vitamin D’s antiviral mechanism has not been fully established. This paper reviews the recent mechanisms by which vitamin D regulates the immune system, both innate and adaptive systems, and reflects on the link between serum vitamin D levels and viral infections.

## 1. Introduction

Viruses are entities that can only replicate within a host cell and can infect a variety of organisms, ranging from single-cell bacteria to mammals. A complete viral particle or virion is made of three main structures: (1) a core containing genetic material (DNA/RNA) packed with basic proteins known as nucleoproteins, (2) a protein coat known as the capsid that encases and protects the genetic material, and (3), in some viruses, a lipid bilayer surrounds the capsid, enveloping the viral capsid [1]. Understanding the structure and the functionality of viruses is crucial for the development of treatments against viral diseases. One of the approaches used to develop treatments is to study how the human immune system reacts to a viral infection.

Studies have shown that Vitamin D (Calciferol) plays a multitude of biological roles in the human body. Vitamin D Receptors (VDRs) can be found in nearly all cells of the body and many sites of the genome [2]. For a century, the link between vitamin D deficiency and susceptibility to various pathogens has been suggested, with the early observation that children with nutritional rickets were more prone to respiratory tract infections, leading to the coining of the phrase “rachitic lung” [3]. In the last two decades, epidemiologic studies have demonstrated strong associations between seasonal variations in vitamin D levels and the incidence of various infectious diseases, which include septic shock [4], respiratory infection [5], and influenza [5,6].

VDR and 1α-hydroxylase, the enzyme necessary for the conversion of vitamin D into its active form, have been found on a variety of immune cells, including circulating mononuclear cells [7,8], indicating that vitamin D serves a significant role in evoking an immune response against the invading pathogen [9]. This discovery has revolutionized the field of vitamin D immunology. Additionally, studies have shown that vitamin D has a critical role in different physiological diseases. Furthermore, vitamin D deficiency can be linked to chronic conditions, including calcium metabolism disorders, cardiovascular disorders, autoimmune disorders as well as cancer, pertaining to both the development and severity of disease [10]. Various studies have been done to link vitamin D with the function of the immune system, specifically those related to autoimmunity. Several studies have shown that vitamin D is involved in the expression regulation of specific endogenous antimicrobial peptides in immune cells [11,12]. In this review, we will discuss the immune-modulatory effects of vitamin D in relation to viral infections and will summarize the recent work done on this topic.

## 2. Vitamin D Synthesis and Immune Modulation

### 2.1. Synthesis of Vitamin D

Vitamin D enters the body through dietary consumption (about 20% of vitamin D3) or is synthesized by the skin (80%) from 7-dihydro-cholesterol after exposure to type B ultraviolet (UVB) which varies seasonally. A representative population study done on the European population showed that about 17.7% of Europeans exhibited vitamin D deficiency during the winter months while the percentage fell to 8.3% during the summer months [13]. The UVB rays from sunlight convert 7-dihydro-cholestrol to pre-vitamin D_3_, which then rapidly converts to vitamin D_3_ (cholecalciferol). After that, vitamin D3 is biologically activated by hydroxylation in the liver carried out by enzymes cytochrome P450 2R1 (CYP2R1) and cytochrome P450 27 (CYP27A1) converting to 25 hydroxyvitamin D3 (25(OH)D3).

The renal 25-hydroxyvitamin D3-1 alpha-hydroxylase (CYP27B1) activates the hydroxylation of 25-(OH)D3 in the mitochondria of kidney tissue to produce 1,25-(OH)2D3 (calcitriol), the active form of vitamin D3. The synthesis and activity CYP27B1 is usually regulated by the fibroblast growth factor 23 (FGF-23) [14]. Once activated, the vitamin D binds to different VDRs on different cells where it can activate transcription of different genes, thus exerting a hormone-like effect [2,15,16]. Other than skin, vitamin D can also be acquired through diet. Foods rich in vitamins D_3_ and D_2_ are processed in the digestive tract; then, the liberated vitamin D is transported to the liver via human serum vitamin D-binding protein (DBP). After this point, the process of conversion follows the previous pathway to the liver [15]. Vitamin D is regulated by the enzyme CYP24A1 that catabolizes excess 25(OH)D3, a metabolite of vitamin D, to calcitroic acid, which is excreted through bile [17].

### 2.2. Vitamin D and Endocrine Function

Vitamin D is a versatile molecule that is involved in many different processes in the human body. As discussed earlier, vitamin D plays an autocrine role where it impacts gene transcription; however, vitamin D also has endocrine functions, the primary being regulation of calcium [9]. Specifically, vitamin D is crucial for the absorption of calcium from food. Calcium can be transported from the digestive tract in a paracellular or transcellular manner. Paracellular movement of calcium is passive, but the transcellular movement is regulated by the activated form of vitamin D which is then transported to the liver [18]. Apart from calcium homeostasis, vitamin D is also involved in other functions such as maintenance of muscle function, prevention from cancer and cardiovascular diseases [18].

### 2.3. Vitamin D and Innate Immunity

Innate immunity is the first line of defense against foreign entities by the immune system. In recent years, studies have indicated that vitamin D also plays an essential role in the immune system, providing protection against several infections as well as regulating the workings of the immune system [19,20,21]. It has been observed that vitamin D plays a critical role in the function of monocyte. The link between monocytes and vitamin D can be made by the enzyme CYP27B1 [22,23]. Monocytes exert their immune reaction by phagocytosing the foreign body and using Toll-Like Receptors (TLRs) and other classes of pattern recognition receptors to recognize the foreign entity. Some gathered evidence suggested that there is amplification in the CYP27B1 activity when this event occurs [10]. This amplification indicates that there is an increase in the locally produced 1,25(OH)2D, which binds to the endogenous VDR and controls gene expression in monocytes [24]. An increase in the transcription of the gene that codes for an antibiotic protein LL37, which is the active form of an antimicrobial peptide named cathelicidin, is observed [11,25,26]. An increase in the level of LL37 shows enhanced function of monocytes, thereby indicating that vitamin D enhances the activity of monocytes, and the insufficiency of vitamin D can have a negative impact on the potency of monocytes [18,27]. Other evidence suggests that Calcitriol could inhibit inflammatory T cell cytokines such as IL-2 and IL-17 and toll-like receptors present on monocytes [28,29]. High doses of calcitriol supplementation in healthy human subjects (1 μg twice per day for 7 days) leads to a dramatic reduction in the levels of pro-inflammatory cytokine IL-6 secreted by peripheral mononuclear cells [30]. All these effects likely combine and result in the induction of potential regulatory T cells, which are important for regulating immune responses and for the development of autoreactivity [31].

In severe infections, the number of innate granulocytic cells, such as neutrophils, increases dramatically. Early studies suggested that neutrophils are the main source of cathelicidin [32]. Recent studies contradicted this finding; one study showed evidence that, despite the expression of VDR on neutrophilic granulocytes, they do not seem to exhibit any CYP27B1 activity that would enable them to convert 25(OH)D into the bioactive form, which is necessary to initiate cathelicidin gene expression [33]. Nevertheless, serum 25(OH)D levels were found to be significantly lower in critically ill septic patients. This can be associated with decreased concentrations of the antimicrobial protein cathelicidin [34]. This finding strengthens the theory that vitamin D status affects antimicrobial protein levels and can be crucial in infection control.

Besides fighting directly against microbes, monocytes and APCs, in particular, dendritic cells (DC), are important targets for the immune-modulatory effects of vitamin D. The APC play an essential role in the activation of the adaptive immune response, as they are involved in the presentation of the foreign antigens to the T and B cells and are capable of regulating them by either immunogenic or tolerogenic signals through cytokines and expression of co-stimulatory molecules [29]. Various studies have demonstrated that vitamin D and its metabolites can alter the function and morphology of DC to induce a more tolerogenic, immature state [19,35,36,37].

### 2.4. Vitamin D and Adaptive Immunity

Early studies investigating the effects of vitamin D on human adaptive immune cells showed that the expression of the nuclear VDR, as well as vitamin D-activating enzymes, was observed in both Band T cells [38]. In T cells, vitamin D acts to reduce the production of cytokine IL-12, an important cytokine for T cell proliferation. Reduction in the production of IL-12 reduces the production of IFN-γ and IL-2. IFN-γ and IL-2 are crucial cytokines for the recruitment of T-cells as well as their proliferation [39]. Vitamin D also inhibits the T cells from differentiating in Th1 and Th17 cells, which exert their effect on intracellular pathogens and are pro-inflammatory [40]. Vitamin D also has an inhibitory effect on B cells. While it is not studied deeply, it is known to inhibit the proliferation of B cells, differentiation of B cells to plasma cells, secretion of immunoglobulins, and formation of memory cells as well as induce apoptosis in B cells [41]. During resting state, VDR expression by the T and B cells is very low; however, upon activation and proliferation, a significant upregulation of VDR expression in T and B cells is observed, allowing regulation of nearly 500 vitamin D responsive genes that affect the differentiation and proliferation of these cells [41,42].

When combined, these results suggest that vitamin D may regulate not only the innate but also the adaptive immune system. Vitamin D supplementation could possibly provide a safe and practical form of therapy in the future to support immune tolerance in autoimmune diseases or following transplantation [42].

## 3. Evidence for Vitamin D Influence on Different Viral Infections

Vitamin D plays an integral role in regulating the immune system. It has been shown in earlier studies that vitamin D levels correspond to the degree of the immune response against viral infections. Vitamin D is known to mediate the innate response by modulating the activity at TLRs, which are common to most immune cells, including macrophages, monocytes, and epithelial cells. Thus, serum levels of vitamin D can be linked to the likelihood of developing and recurrence of viral infections.

### 3.1. Vitamin D and Rhinovirus

Human Rhinovirus (HRV) is the most common cause of upper respiratory tract infection (URI), which, in turn, is the most frequently occurring acute infection in the industrialized world, with estimates of about two to three episodes per year in adults, and five to seven episodes per year in children [43,44]. HRV infection leads to considerable economic burdens in terms of medical visits and school and works absenteeism [45,46]. Nevertheless, only one study has investigated the effect of vitamin D on rhinovirus infection [47]. Schneider et al. treated primary human bronchial epithelial cells (hBECs) with 25(OH)D or 1,25(OH)2D and subsequently infected the cells with RV-16. They reported that incubation of the cells with vitamin D metabolites did not effect on RV-16 replication. However, incubation with vitamin D enhanced the secretion of pro-inflammatory chemokines CXCL8 and CXCL10, both in the presence and absence of rhinovirus infection. These chemokines are involved in the recruitment of immune cells such as macrophages, neutrophils, and T-cells to the infection site, and thus could act as an effector mechanism as to how vitamin D alters the antiviral response to HRV infection.

### 3.2. Vitamin D and Influenza Viruses

One of the most common types of viral infections are those that affect the respiratory system. Respiratory viral infections lay a massive burden as they are one of the main causes of infant death across the globe. One of the approaches to prevent this problem is to vaccinate infants; however, this is only successful in 60% of all cases, depending on the influenza strain [48,49]. Not only infants but adults also bear the burden of the disease. Influenza can be lethal for the elderly, adults with co-morbidities, and obese adults [50]. Since influenza infections can have detrimental consequences and the vaccine does not have a satisfactory success rate, researchers are now looking at other preventative measures that can be taken against influenza infections.

Previous work on adults who are affected by influenza viruses has shown that they also exhibit a higher incidence of hypovitaminosis due to factors that affect vitamin D metabolism such as the efficiency of skin at synthesizing vitamin D or reduced renal production of the active form of vitamin D, 1,25(OH)2D [50]. An observational study done between 1980 and 2000 in Norway found that influenza-related mortality during the colder season was linked to the low level of vitamin D among people [51]. An observational study conducted on school children investigated the effects of vitamin D supplementation in the prevention of influenza A infection during the winter season when the incidence of flu is much higher. They found that, in the group of children who received a placebo, the number of children infected (18.6%) was almost twice as high as in the group of children who had received the actual supplementation (10.8%) [13].

While there are studies that demonstrate that restoration of normal vitamin D levels can provide better protection against influenza virus infection, a significant number of studies suggest that there is no correlation between vitamin D levels and the prevention of viral infections in the case of influenza virus. A study was carried out on 1641 children with influenza infection, where 177 children were given vitamin D supplements (14,000 IU/wk for eight months), and 209 children were given a placebo. This study is done regarding the correlation between vitamin D and respiratory infections, concluding that vitamin D had reduced non-influenza respiratory viral infection; however, no decrease had been documented with influenza respiratory viral infection [49]. To study the effect of vitamin D on innate immune response against the influenza strain H9N2, found commonly in turkeys, research was conducted on epithelial cells, which showed that treatment using calcitriol has negative effects on the immune response against the H9N2 virus, specifically during the later stage of the disease. An increase in the activity of VDR due to high levels of calcitriol results in pro-inflammatory response, which has adverse effects on the disease stage [52]. Another study suggests that vitamin D supplementation did not show help in preventing the influenza infection and rather, surprisingly found that supplementation made the disease last longer compared to duration observed in patients who received a placebo [53].

Some studies have also looked into the effect of vitamin D on the efficacy of immunization against influenza viruses. Some works of research have shown a positive link between vitamin D levels and the effectiveness of vaccines. A study conducted on the efficacy of vaccines based on vitamin D levels, and it was found that the depletion of vitamin D to normal levels shows a better immunological response to the vaccine [54]. On the other hand, some studies have shown that vitamin D supplementation can reduce the efficacy of a vaccine. A study performed on 135 children, who were vaccinated with live attenuated influenza vaccine (LAIV) or inactivated influenza vaccine (IIV), showed that lower levels of vitamin D demonstrated better response to the vaccines [55]

Considering all the above-detailed studies, there are mixed findings that correlate vitamin D levels with prevention from influenza infections. Some demonstrate that vitamin D levels are helpful against influenza infections while others state that there are no effects and, in some cases, a negative correlation is seen. This indicates that there might be other mechanisms involved in regulating the immune response against influenza infections that are yet to be studied.

### 3.3. Vitamin D and Respiratory Syncytial Virus

Respiratory syncytial virus (RSV) is the major cause of bronchiolitis in one-year-old infants, with nearly all children exhibiting serologic evidence for infection with the virus by the age of 2–3 years [56,57]. Studies have demonstrated that vitamin D deficiency can be linked to RSV susceptibility, with low concentrations of 25(OH)D in cord blood plasma related to RSV incidence in the first year of life [58]. In addition, studies have shown that single-nucleotide polymorphisms in the VDR vitamin D-binding protein can be associated with a genetic predisposition to RSV bronchiolitis [58,59].

On the other hand, Hansdottir et al. have shown that infecting human tracheobronchial epithelial (hTBE) cells with RSV decreases IκBα, due to RSV-induced degradation, whilst, treating the cell with 1,25(OH)2D before RSV infection, increases the expression of the NF-κB inhibitor IκBα [60]. In addition, they showed that the increase in IκBα results in NF-κB remaining inactive in the cell, which, in turn, prevents the translocation of NF-κB into the nucleus and subsequent binding to DNA promoter regions, which will eventually lead to inhibition of IFN-β and CXCL10 secretion which is driven by NF-κB. Furthermore, Hansdottir and his colleagues showed evidence that treatment with 1,25(OH)2D resulted in decreased levels of human myxovirus resistance protein 1 (MxA) and interferon-stimulated genes 15. Even though, treating the hTBE cells with 1,25(OH)2D reduced the production of important antiviral components responding to RSV infection (IFN-β, CXCL10, MxA, and ISG15), viral replication and viral load did not increase. Thus, through increasing IκBα levels, vitamin D is able to decrease the inflammatory response to RSV infection while maintaining the antiviral state and without having adverse effects on viral load. This suggests that vitamin D plays a potential role in reducing immunopathology, which is essential in reducing the disease severity and consequently in reducing the morbidity and mortality from this common infection.

Studies have shown that antimicrobial peptides LL37 and β-defensin 2 have antiviral activity against RSV infection via blocking the cellular entry of the virus and preventing the epithelial cell death induced by the virus, which leads to inhibiting the production of new infectious particles and, ultimately, diminishing the spread of the virus in the host [61,62].

### 3.4. Vitamin D and Dengue Virus

Dengue virus (DENV) is a highly endemic infectious disease of tropical countries and is rapidly becoming a global burden. Dengue fever is a mosquito-borne tropical disease caused by the dengue virus. Severe dengue disease can be linked with high viral loads and overproduction of pro-inflammatory cytokines, suggesting dysregulation in the mechanisms that modulate cytokine production throughout the infection. Controlling the secretion of pro-inflammatory cytokines by macrophages are essential events needed to avoid the progression of dengue disease. A study has shown that human monocyte-derived macrophages differentiation in the presence of vitamin D restricts DENV infection by influencing DENV binding to cells [63]. In addition, they showed that secretion of TNF-α, IL-1β, and IL-10 was significantly lower in DENV-infected MDMs treated with vitamin D than in MDMs without vitamin D treatments [63]. Studies have shown evidence that vitamin D can control cytokine responses by indirect regulation of NF-κB activity or by direct modulation of the VDR-dependent gene [60,62,63].

### 3.5. Vitamin D and Hepatitis C Virus

One of the leading causes of liver-related mortality is an infection caused by the hepatitis C virus. HCV is a single-stranded, enveloped virus belonging to the *Flaviviridae* family. In many adults, the infection caused by the virus is asymptomatic. If the infection does not resolve on its own, the infection can become chronic and cause irreversible liver damage [64]. Due to this dangerous nature of the virus and the poor outcome of the disease on the patient’s life, researchers are now looking at improved ways to prevent and treat the infection.

As vitamin D is known to be an immunoregulator, studies have been done to correlate the levels of vitamin D with the occurrence of vitamin D infection. Patients with liver disease are commonly accompanied by Vitamin D deficiency; however, the mechanism behind this occurrence is yet to be understood [65]. 25-OH vitamin D has multiple effects, acting as an innate antiviral agent and also exhibits an anti-fibrotic effect, simultaneously capable of producing more than one benefit. In addition, vitamin D is a powerful immune-modulator known to play a significant role in inflammatory responses as well as fibrosis caused by HCV infection [48,66,67].

Vitamin deficiency is commonly observed in patients with chronic hepatitis C infection. While it is known that HCV infection can reduce the level of vitamin D in patients, it has been speculated that vitamin D levels before acquiring the disease and during infection can affect the disease outcome. One study conducted on immuno-competent patients with recurrent hepatitis c infection showed that the administration of vitamin D with antiviral therapy increased the likelihood of sustained viral response (SVR) [68]. This finding has been backed up by other studies as well. One study found that supplementation of vitamin D with Peg-α-2b/ribavirin therapy will improve the immune response against HCV genotype 1 [69]. Another study found that adding vitamin D to conventional Peg/RBV therapy for patients with HCV genotype 2–3 significantly improves viral response [70]. Along with decreased probability of SVR, low vitamin D levels have also been correlated with severe fibrosis. It has been observed that the interaction of vitamin D with vitamin D receptors in fibroblasts can protect the cells from oxidative damage, can affect proliferation, gene expression, and migration of fibroblasts as well as reduces the inflammatory activity of liver stellate cells [71]. However, some studies suggest that vitamin D supplementation does not produce any significant improvements in disease outcome. A study found that vitamin D supplementation given with antiviral therapy for patients affected by HCV genotype 1 did not show any difference in efficacy of the treatment or produce any beneficial effects in terms of fibrosis progression [48].

Vitamin D can also be used as a prognostic factor in cirrhosis in HCV affected patients. Cut-off levels of less 7.2 ng/mL, which is related to high mortality rates, lower than 7.1 and 6.6 ng/mL predict mortality in SBP (spontaneous bacterial peritonitis) and HE (hepatic encephalopathy) patients, respectively [72]. HCV-related cirrhosis could probably lead to vitamin D deficiency included malabsorption to portal hypertension and hydrolyzation of 25-OH vitamin D [72]. However, when using vitamin D levels as a prognostic tool, certain factors should be considered. People who live in the Middle-East tend to exhibit lower vitamin D levels than people from other geographical areas due to factors such as darker skin color, consumption of diet poor in vitamin D, and others [73]. It is wise to consider the effects of race and other factors, such as sun exposure and vitamin D supplements intake. It has been established that racial differences affect the 25-OH vitamin D concentrations, parathyroid hormone (PTH), and calcium homeostasis. The racial study, however, shows some limitations such as individual sun exposure, ancestry profile, and possible bias of measured vitamin D due to the time of the year the samples are collected [65,74]. It has been known that people are more exposed to light during the summer months compared to the winter season, so this could give out false results that is agreeable to its suspected to the season the samples were taken.

Most of the findings on this subject suggest that vitamin D supplementation with antiviral therapy can improve the outcomes of the disease and prevents recurrence of HCV infection.

### 3.6. Vitamin D and HIV

The effect of vitamin D on the development of HIV infection and its prevention is still not known and needs to be studied; however, some researches have demonstrated that vitamin D levels in HIV patients can be used as prognostic markers. A study performed on untreated African women showed that a low vitamin D level was associated with poor prognosis of the disease [75]. Again, factors such as race, sunlight exposure, weather, genetics, and others should be considered when using vitamin D as a prognostic tool since these factors can affect the level of vitamin D within populations.

Some studies also suggest that vitamin supplementation with viral therapy can improve the outcome of the disease. One study suggests that vitamin D deficiency is associated with higher rates of mortality. This can be due to the role vitamin D plays in regulating the immune system. Vitamin D induces the expression of antimicrobial peptides such as cathelicidin and defensin β2 and, when the levels of vitamin D fall below 20 ng/mL, the cathelicidin response is not initiated, leading to further impairment of immune system and subsequent increase in opportunistic infections [76]. Another study showed evidence that vitamin D supplementation hindered the disease progression. As discussed earlier, vitamin D is involved in regulating the CD4 function and sufficient levels of vitamin D prevent viral entry into the cells and cytopathogenic effects [77].

Vitamin D supplementation can also be used to improve bone mineral density (BMD) and strengthen the bones to prevent fracture prevention [78,79,80]. It is more important to check and characterize the vitamin D status in HIV-infected children because the peak bone mass is not reached until the age of 25 [78]. Although vitamin D is relevant in people suffering from osteoporosis in African American people, the race is definitely a factor among HIV-infected people [81]. There are other factors associated with low vitamin D levels in HIV-infected people, and this includes high body mass index (BMI) and obesity [78].

In the studies discussed above, it can be inferred that vitamin D levels are indicative of the prognosis of HIV infection, and vitamin D supplementation can improve the outcome of the disease.

### 3.7. Vitamin D and SARS-CoV-2

As of late 2019, the globe is facing a new pandemic named Covid-19. This pandemic originated in the Wuhan district of China and has rapidly disseminated all across the world. The causative agent behind this disease is SARS-CoV-2. This virus belongs to a large family of viruses known as *Coronaviridae*. Based on phylogenetic and genomic links, the viruses in this family can be divided into four genera: alphacoronaviruses, betacoronaviruses, gammacoronaviruses, and deltacoronaviruses [82]. Like SARS-CoV and MERS-CoV that have caused epidemics previously, SARS-CoV-2 is also a betacoronavirus and consists of a non-segmented, single-stranded, positive-sense RNA that closely matches its predecessor, SARS-CoV [83]. According to the genetic analysis of the viral genome done on patients in China, there are two types of SARS-Cov-2 viruses: L-type and S-type. The L-type is more common and infectious than the S-type, which is the ancestral version of the genome [84]. The genome of SARS-CoV-2 virus codes for four proteins: spike (S), envelope (E), membrane (M), and nucleocapsid (N) proteins, all of which are essential for viral infectivity [85]. The S (spike) protein, is relatively the most crucial in infecting the host. This protein is present on the surface of the virus, and it interacts with the ACE2 receptor of the host cell. Upon interaction, the proteases such as transmembrane protease serine 2 (TMPRSS2) cleave off the C-terminal of ACE2, which mediate membrane fusion between the viral particle and the host cell membrane, allowing the virus to enter the cell and begin replication [86,87,88].

SARS-CoV-2 is associated with severe respiratory disease and multi-organ failure; however, compared to other zoonotic-origin coronaviruses that cause infections in humans, such as the SARS-CoV from 2003, it is less lethal. A majority of people infected by the virus do not show any symptoms and, out of those who do present symptoms, only a small percentage is severely affected. This variability in symptoms can be attributed to the difference in immune response among individuals. Based on clinical findings, when the virus enters the body, it undergoes a non-severe incubation stage. During this stage, if the immune system of the host elicits a specific immune response, the virus can be eliminated before the disease progresses. This ability to eliminate the virus during the initial stage of the disease is determined by the immune status of the individual (age, health condition, HLA type, ABO blood group, and others). If the host cannot eliminate the virus at the initial stage, the virus will attack tissues with high ACE2 receptor expressions such as the lungs and the kidneys. This will activate inflammation mediated by the innate immune system, specifically by pro-inflammatory cytokines such as (IL)-1B and IL-18 released by macrophages and type 1 T helper (Th1) released by immune cells, that will cause further damage to the organs and lead to the severe stage of the disease [89,90].

As vitamin D plays a crucial role in regulating the immune system and is known to be protective against viral infections, it is suspected that vitamin D levels may correspond to the likelihood of SARS-CoV-2 infection and the outcome of Covid-19. The areas where there is a higher incidence of Covid-19 are located on higher latitudes and have low exposure to sunlight. This could correlate to lower levels of vitamin D, which could increase the susceptibility to Covid-19 [91]. It has also been found that people who are categorized as highly susceptible to Covid-19 also exhibit vitamin D deficiency. Those afflicted by chronic diseases such as diabetes mellitus and cardiovascular diseases, obese and older population are considered a high-risk group to Covid-19 and, in most cases, are vitamin D deficient [92,93].

In the lungs, type II pneumocytes are the main target of SARS-CoV-2 and are involved in surfactant production in the alveoli [94]. Upon infection, these cells are destroyed by the severe immune response, which reduces the surfactant production in the lungs and often leads to complications in patients affected by Covid-19. Studies have shown that metabolites of 1,25-dihydroxy vitamin D stimulate the production of surfactants in type II pneumocytes and can reduce the surface tension caused by surfactant depletion in Covid-19 patients [95]. Most complications caused by Covid-19 occur due to the severe immune response that damages integral organs. It has been found that vitamin D prevents excessive release of pro-inflammatory cytokines and chemokines by modulating the activity of macrophages and thereby preventing tissue damage [96].

Currently, data that directly correlate the effect of vitamin D on the susceptibility and severity of Covid-19 are lacking, but since vitamin D has multiple effects on the immune system, it is likely to be involved in modulating the immune response against SARS-CoV-2.

## 4. Conclusions and Future Directions

Vitamin D has immuno-modulating properties that affect both innate and adaptive immunity (Figure 1). As vitamin D plays an integral role in regulating the immune system, researchers have been studying the role of vitamin D disease mechanisms (Table 1). Our review compiles the recent findings on the work done on relating the levels of vitamin D with different viral infections outcomes. We found that only few viruses have been studied in relation to the effect vitamin D has on them. While most of the work has been done on chronic infections such as HCV and HIV, there is some substantial research done on the influenza and other viruses. Until now, there has been no direct correlation made to vitamin D levels and Covid-19 outcome. Other viruses have not been studied in depth. However, as this paper shows that vitamin D affects the immune system in various ways, future studies are needed to understand the role of vitamin D in preventing infections caused by various viruses as well as how re- stabilizing the vitamin D levels in diseased patients can improve the outcome of the disease. Some of the works mentioned above have shown some promising results and if experimentation with other viruses shows similar results, vitamin D can be used as a readily available and inexpensive form of adjunct therapy. This will help improve the outcome of the disease and the quality of life of patients.

## Figures and Tables

**Figure 1 nutrients-12-02879-f001:**
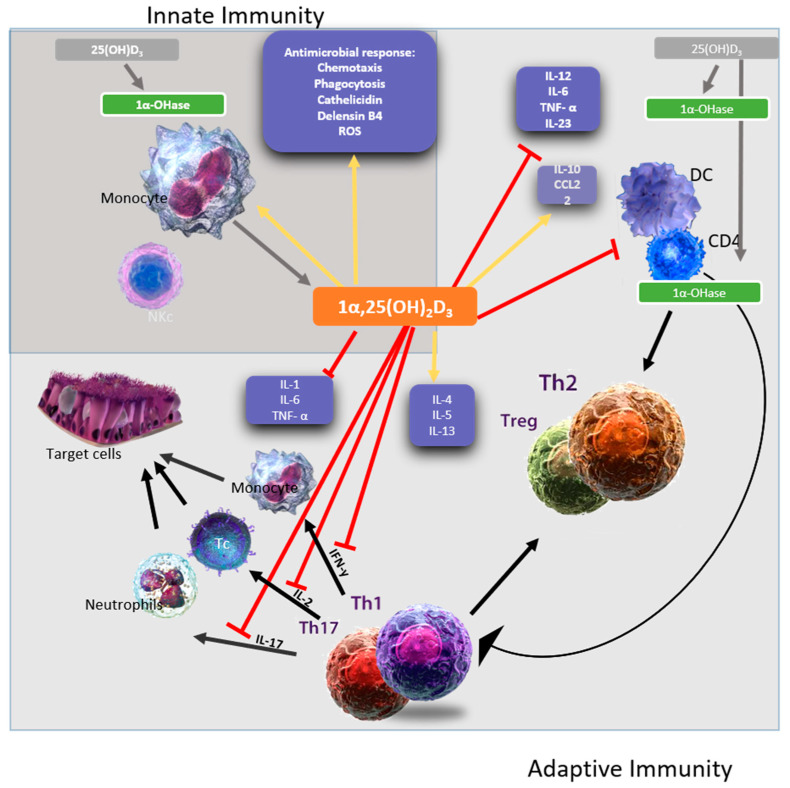
A representative figure of the immunomodulatory effects of 1,25(OH)2D3. 1,25(OH)2D3 targets different players of the innate (dark grey square) and adaptive immune (light grey square) compartment. Vitamin D has been shown to enhance chemotaxis, antimicrobial peptides, and macrophage differentiation to stimulates the innate immune responses. In addition, Vitamin D also stimulates the adaptive immune responses. For example, at the level of the antigen-presenting cells, like dendritic cells, Vitamin D inhibits the surface expression of the MHC-II-complexed antigen, co-stimulatory molecules, and the production of IL-12 and IL-23 cytokines leading to indirectly shifting the polarization of T cells from a Th1 and Th17 phenotype towards a Th2 phenotype. The figure is adapted from [97,98].

**Table 1 nutrients-12-02879-t001:** Latest articles describe the immune modulatory effect of Vitamin D on viral infection.

Virus	Study Date	Type of Study	Age	Study Sample	Assay	Vitamin D Type	Duration	Effect of Vitamin D	Clinical Marker	Source of Immunity	Proposed Mechanism	Ref
HIV	2019	Randomized, double-blind, placebo-controlled, clinical trial	>18 years	52 HIV negative control, 173 patients completed the study	HIV viral load assay BD FACS counter CLIA	25(OH)D3	16 weeks	No significant change	HIV viral load	-	-	[2]
Influenza A	2018	multicenter, randomized, open, controlled clinical trial	3−12 months old	400 infants	ELISA	25(OH)D3	4 months	High dose of Vit D, better prevention	Viral Load Detection of Influenza Virus A	Immunomodulatory effects	Antiviral peptides	[3]
HIV	2018	randomized, active-control, double-blind trial	8–25-year-old	51 samples	Automated chemiluminescent technique	25(OH)D3	12-month	Vitamin D supplementation decreased markers of T-cell activation/exhaustion and monocyte activation	CD4+ T-cell count, CD8+ T-cell count, or CD4/CD8 ratio, monocyte subsets	-	CD4/CD8 activation	[4]
Upper respiratory infections: influenza A and B, adenoviruses, RSV, picornaviruses, coronavirus, human metapneumovirus, and parainfluenza viruses	2017	randomized clinical trial	1 to 5 year	703 participants	LuminexxMAP ID-Tag RVP assay system	25(OH)D3	4 months	vitamin D supplementation did not reduce overall wintertime upper respiratory tract infections (compared between 400 IU and 2000 IU)	presence of respiratory viruses	-	-	[5]
HCV	2017	randomized, double-blind, placebo-controlled, interventional study	18 and 70 years	80 patients (40 samples, 40 placebo)	Liaison 25 OH vitamin D total assay	Vit D2	6 weeks	Higher vitamin D level, better treatment outcome	HCV viral load, T-helper1/2 cytokines, IP-10 and DPP IV levels as compared to placebo		Lower serum IP-10 and DPP IV levels	[6]
HCV	2016	two large clinical trials	No age restriction	1292 patients	DiaSorin LIAISON 25(OH)D TOTAL assay	25(OH)D3	12 weeks	No beneficial response	HCV RNA level, Liver markers	-	-	[7]
RSV, influenza and other community acquired pathogens	2016	Randomized placebo-controlled dose-ranging trial	>18	1300 pregnant women	polymerase chain reaction (PCR) (for virus)	25(OH)D3	0–6 months postpartum	Prenatal Vit D prevents Acute RI in infants	8 respiratory viruses	-	-	[8]
URT (not specified)	2015	Randomized placebo-controlled	No age restriction	82 swimmers	RT PCR (for viruses)	25(OH)D3	12 weeks	No link between thymus activity and Vit D levels	T cell receptor excision circles (markers of thymus activity)	-	-	[9]
HCV	2011	intention-to-treat prospective randomized study	18–65 years	72 patients	RT PCR (for viruses)	25(OH)D3	48 weeks	High vitamin D levels improve immune response	Plasma HCV-RNA	T cell function modulation	Better TH2 and Treg function	[10]

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
