# Peer review of "Immune Modulatory Effects of Vitamin D on Viral Infections"

_nutrients, 2020, doi:10.3390/nu12092879_

Round 1

Reviewer 1 Report

The Title should be changed to “Immune-modulatory Effects of Vitamin D on Viral Infections”.

Abstract

Recent evidences indicated that vitamin D also plays an essential role in the immune response against viral infections and suggested that deficiency in vitamin D leads to increased susceptibility to viral infections as well as higher chances of recurrence.

Please change to “…suggested that vitamin D deficiency increases susceptibility to viral infections as well as the risk of recurrent infections.

Introduction

L32 Understanding the structure and the functionality of viruses has been crucial to the development of

Please change to “Understanding the structure and the functionality of viruses is crucial for the development of…”

When abbreviating a term, use the full term the first time the authors use it, followed immediately by the abbreviation.

L59 enzyme DHCR7 --> Please provide the full name of DHCR7.

L63 25(OH)D3 --> Please provide the full name of 25(OH)D3 and 1,25(OH)2D3.

L 64  Then, the 1,25(OH)2D3 (fully active metabolite) is hydroxylated in the kidney by the enzyme CYP27B1 parathormone (PTH), and the fibroblast growth factor 23 (FGF-23) control CYP27B1 synthesis and activity.

Please clarify this part.

If the authors would like to focus on immune-modulatory effects of vitamin D on viral infections, it was not necessary to include 1.2. Vitamin D and Endocrine Function (L 73-82) and 2. Vitamin D and Autoimmunity (L 135-149).

Structure consistency for each viral disease is suggested. Such as Pathogenesis, XXX virus and Host Interaction, Innate and Adaptive Immune Response in XXX Infection, vitamin D and XXX viral infection, …

L79 vitamin D (1,25-dihydroxyvitamin D3); vitamin D , vitamin D3 (1,25-dihydroxyvitamin D3)

Please do not use these terms interchangeably.

L 291 SBP, HE --> Please provide the full name.

Author Response

Dear editor,

The authors of this paper would like to sincerely thank you and the reviewers for the detailed analysis of the paper as well as for their helpful comments. All the improvements suggested by the reviewers have been made. All changes in the manuscript are highlighted yellow. Shall you have any more concerns, kindly contact the corresponding author at [email protected].

………………………………………………………………………………………………………………………

Responses to reviewers’ comments

Revvirwer1

Abstract

Recent evidences indicated that vitamin D also plays an essential role in the immune response against viral infections and suggested that deficiency in vitamin D leads to increased susceptibility to viral infections as well as higher chances of recurrence.

Please change to “…suggested that vitamin D deficiency increases susceptibility to viral infections as well as the risk of recurrent infections.

> Change is made as suggested.

Introduction

L32 Understanding the structure and the functionality of viruses has been crucial to the development of

Please change to “Understanding the structure and the functionality of viruses is crucial for the development of…”

> Change is made as suggested.

When abbreviating a termuse the full term the first time the authors use it, followed immediately by the abbreviation.

L59 enzyme DHCR7 --> Please provide the full name of DHCR7.

> Does not appear in the text anymore.

L63 25(OH)D3 --> Please provide the full name of 25(OH)D3 and 1,25(OH)2D3.

> Changes are made as suggested.

L 64  Then, the 1,25(OH)2D3 (fully active metabolite) is hydroxylated in the kidney by the enzyme CYP27B1 parathormone (PTH), and the fibroblast growth factor 23 (FGF-23) control CYP27B1 synthesis and activity.

Please clarify this part.

> Changes were made to the whole paragraph to enhance clarity of the sentence.

If the authors would like to focus on immune-modulatory effects of vitamin D on viral infections, it was not necessary to include 1.2. Vitamin D and Endocrine Function (L 73-82) and 2. Vitamin D and Autoimmunity (L 135-149).

> Considering the interplay between difference systems in the human body, including the endocrine and immune system, we believe that the added paragraph, even short, could of interest to some readers.

Structure consistency for each viral disease is suggested. Such as Pathogenesis, XXX virus and Host Interaction, Innate and Adaptive Immune Response in XXX Infection, vitamin D and XXX viral infection, …

L79 vitamin D (1,25-dihydroxyvitamin D3); vitamin D , vitamin D3 (1,25-dihydroxyvitamin D3)

Please do not use these terms interchangeably.

> We restricted the used of these terms in paragraphs 1.1 (synthesis of vit D) and 1.2 ( Vitamin D and Endocrine Function). Otherwise, the term Vitamin D was used in the rest of the review.

L 291 SBP, HE --> Please provide the full name.

> Full names are now provided. 

Reviewer 2 Report

This review is quite thorough, although there are a number of issues with the contents. An annotated version of the manuscript is attached with several comments and corrections to the English.

Major concerns:

  1. The depiction of vitamin D synthesis in Fig. 1A in the context of a review on vitamin D and immunity is misleading as most of the 1,25OH2D produced in the immune system is likely produced locally in response to infection.
  2. The section on vitamin D and autoimmunity is not necessary and should be removed.
  3. The reference list is incomplete for two reasons. First, some key references are not cited at all. Also, about 25 references cited in the text don't appear in the reference list.
  4. The section describing sources of vitamin (20% for diet and 80% from the sun) is incomplete and misleading.

Author Response

Major concerns:

  1. The depiction of vitamin D synthesis in Fig. 1A in the context of a review on vitamin D and immunity is misleading as most of the 1,25OH2D produced in the immune system is likely produced locally in response to infection.

> We thank the reviewer for his comment. Figure 1A is now deleted.

  1. The section on vitamin D and autoimmunity is not necessary and should be removed.

> Many viral infection are known to induce autoimmunity. Considering the interplay between viral infections and vitamin D, we believe that the section on vitamin D and autoimmunity would be of interest to the reader.

  1. The reference list is incomplete for two reasons. First, some key references are not cited at all. Also, about 25 references cited in the text don't appear in the reference list.

> Reference list is now updated.

  1. The section describing sources of vitamin (20% for diet and 80% from the sun) is incomplete and misleading.

> We have added more information into section 1.1 (synthesis of vitmin D) and we hope it suffices.

Round 2

Reviewer 2 Report

The manuscript is substantially improved. However, I note that reference 10, which should be references 10 and 11 as there are two incomplete references in one, is incomplete.

Author Response

The authors have provided a substantially improved version of their manuscript.
Two issues still should be corrected:
1. ) Titel: at present, it seems like vitamin D only is addressed in the review, Please modify the title accordingly.

Response: Vitamin D Has been added to the paper. 

2.) The depiction of vitamin D synthesis in Fig. 1A is misleading as most of the 1,25OH2D produced in the immune system is likely produced locally in response to infection. Please indicate the local 1,25OH2D biosynthesis in the figure and mention it in the legend.

Response: Figure 1A had been removed earlier as per one of the reviewer's suggestions. 
..............................................
Comments and Suggestions for Authors

The manuscript is substantially improved. However, I note that reference 10, which should be references 10 and 11 as there are two incomplete references in one, is incomplete.

Response: References have been fixed. 
